# Content Validity and Utility of the Collaborative Process for Action Plans to Achieve Children's Participation Goals

**Robert J. Palisano** [1,*], **Lisa A. Chiarello** [1] , **Nea Vänskä** [2] **and Salla Sipari** [3]

1    Department of Physical Therapy and Rehabilitation Sciences, Drexel University, Philadelphia, PA 19102, USA
2    School of Rehabilitation and Examination, Metropolia University of Applied Sciences, 00920 Helsinki, Finland
3    School of Wellbeing, Metropolia University of Applied Sciences, 00920 Helsinki, Finland
*    Correspondence: robert.j.palisano@drexel.edu

**Abstract:** Content validity and clinical utility of the Collaborative Process for Action Plans to Achieve Children's Participation Goals were evaluated. The collaborative process is designed to assess child, family, and environment strengths and areas for improvement specific to a child's participation goal and identify intervention strategies and the person(s) responsible for each strategy. Twelve pediatric therapists participated in one of two Nominal Group Consensus Process. Following discussion, therapists rated the importance of child, family, and environment attributes, clarity of wording, and the usefulness of the collaborative process. Ratings for 91% (first group consensus) and 100% (second group consensus) of the statements met the criterion for consensus, supporting content validity. Ten parent/child/therapist teams evaluated clinical utility. Written responses to open-ended questions were coded using inductive content analysis. Parents and therapists indicated that the collaborative process promotes engagement, the goal is considered from many viewpoints, and there is a joint commitment to the action plan. Limitations included time to complete, the need for preparation and more guidance, and unfamiliar expressions. Familiarity with collaborative, solution focused processes and participation interventions are considerations for use in practice. Research is recommended in which the action plan is implemented, progressed, and achievement of participation goals are evaluated.

**Keywords:** action plan; collaborative process; clinical utility; content validity; participation; pediatric rehabilitation

## 1. Introduction

Participation, broadly defined as involvement in life situations [1] is an important outcome of pediatric rehabilitation. Through participation children form friendships, learn skills, express creativity, develop positive identities and emotional well-being, and become self-determined [2,3]. Participation is complex, multi-dimensional, individualized, and characterized by person-environment interaction. Participation includes attendance ("being there") and involvement ("the experience") in meaningful life activities [4]. Consequently, many considerations are necessary for developing action plans (also referred to as intervention plans or care plans) to achieve participation goals. Within a family centered approach to pediatric rehabilitation, these considerations are addressed through collaboration between family members and service providers [5].

Family-service provider collaboration refers to a partnership in which the family and service providers work together to make informed decisions about services and supports [6]. Family-service provider collaboration is characterized by two complementary processes: relational practices (e.g., showing respect and empathy, active listening) and participatory practices (e.g., engaging the family in the intervention process, and incorporating family needs and priorities into intervention) [7–9]. Blue-Banning et al. [10] conducted focus groups and individual interviews with parents of children with and without disabilities, service providers, and administrators and identified six relational practices that

promote family-service provider collaboration: communication, commitment, equality, skills/competence, trust, and respect. Parents emphasized the importance of frequent, open, and honest communication. They voiced that service providers should recognize the importance of their relationship with parents and children, acknowledge parents' points of view, and not be afraid to admit when they do not know something. Service providers spoke of the importance of accepting a family "where they are" and displaying a nonjudgmental attitude toward the family. King et al. [11] emphasize the importance of engaging children and parents in the rehabilitation process in a way that generates value by affective relationship, working together, and supporting motivation and feeling of being valued.

Participatory practices are implemented less often and may be more difficult than relational practices [8,12,13]. Collaborating with families to address their needs and concerns was perceived by both families and service providers as an area for improvement [8,14]. Identifying the role of the family, engaging the family in the intervention process, and providing services that address child and family needs have been identified as challenging participatory practices [15–17]. Piskur and colleagues [18] in a qualitative study of parent lived experiences in enabling participation of their children with physical disabilities advocated for the active use of parents' knowledge and experiences to inform pediatric rehabilitation.

Goal setting is the participatory practice that has been studied most often. A scoping review of 62 studies by Pritchard-Wiart and Phelan [19] identified processes and outcomes related to goal setting in children with motor disabilities. The role of parents was most often described as collaborating with therapists using individualized measures such as the Canadian Occupational Performance Measure [20] and Goal Attainment Scaling [21]. Collaborative processes, however, were unspecified or not well described. Similarly, the role of the child was unspecified or not well described. Only four studies described ongoing processes such as monitoring goal progress and only six studies directly addressed therapy activities to achieve goals.

An et al. [22] proposed a collaborative model that incorporates specific participatory practices to engage families in collaborative goal setting including: the client-centered interview process of the Canadian Occupational Performance Measure [20], visualizing a preferred future and scaling questions of Solution-Focused Therapy [23–25], and the family routine and activity matrix [26]. A randomized control trial found that physical therapists who received instruction on implementation of the strategies interacted more with parents during therapy sessions compared with therapists in the comparison group [27]. Parents in the experimental group were more confident in carrying out activities during daily routines and worked together with therapists to a greater extent than parents in the comparison group. Therapists in the experimental group perceived that they provided information/instruction and worked together with parents to a greater extent than therapists in the comparison group [28].

Although interventions to optimize participation of children with disabilities such as Occupational Performance Coaching [29]; Context Therapy [30]; Environmental Based Intervention [31,32]; Pathways and Resources for Engagement and Participation [33], and Adapted Community Events [34] advocate family-therapist collaboration, specific participatory practices for development of action plans to achieve participation goals are not provided. Palisano et al. [35] have proposed a model for participation-based therapy centered on the premise that achievement of goals for participation at home, school, and in the community are optimized through a collaborative process in which information is shared, solutions to challenges are identified, and action plans are implemented that build on strengths of the child, family, and environment. As illustrated in Figure 1, the collaborative process is based on the perspective that meaningful participation is determined by the dynamic interaction of attributes of the child, family, and environment. The conceptual framework aligns with others [33,36] but the translation to practice can be challenging.

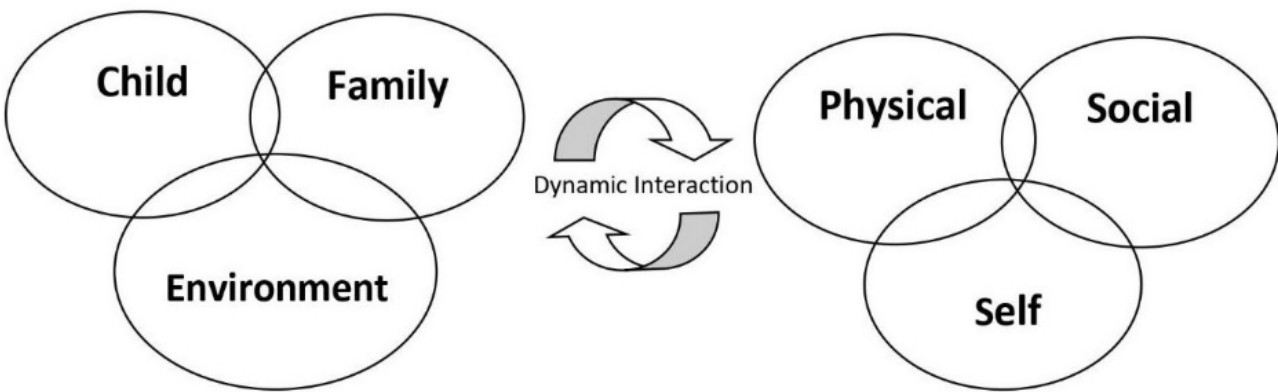

**Figure 1.** Conceptual framework of meaningful participation of children with physical disabilities. Reprinted with permission from Ref. [35]. Copyright 2012 Disability and Rehabilitation.

To address translation to practice, the Collaborative Process for Action Plans to Achieve Children's Participation Goals (referred to in this article as the Collaborative Process for Action Plans) was developed to provide a systematic but flexible process to guide families and therapists through the many considerations necessary to develop action plans [35]. The Collaborative Process for Action Plans provides suggested guiding questions to engage the child, family, and others in conversations for planning an intervention to achieve a participation goal. This includes questions to identify current abilities and considerations, development of intermediate objectives (what needs to happen to achieve the participation goal), as well as then integrating information to develop the action plan for the participation goal. The process is adaptable for client and practice contexts and compatible with contemporary approaches to children's participation.

The form for the Collaborative Process for Action Plans is presented in Supplemental Materials (File S1). Information about the child including any medical restrictions, precautions, and safety concerns, the family member(s) and service provider(s) completing the assessment process, and the participation goal is recorded at the top of the form. Child, family, and environment attributes specific to the participation goal are qualities or characteristics associated with learning and participation such as a child's interest in an activity, family support of the child's participation, and a community program that provides accommodations for children with activity limitations (Table 1). The attributes provide a framework to assess the child's interests and abilities, the family's situation, availability and accessibility of the desired activity, and to identify solutions or what needs to happen (outcomes) to achieve the participation goal (Figure 2).

The Collaborative Process for Action Plans is flexible. Some attributes listed on the form may not apply for certain goals; for other goals, attributes not listed might be important. After the child, family, and therapist identify strengths and outcomes needed to achieve the participation goal, the time and effort that might be needed to achieve the goal are discussed. Finally, whether the goal is the right match for the child and family at a particular time is considered. If yes, the action plan is developed. If no, the goal is modified. In an effort to address how to optimize collaboration, examples of open-ended questions are provided to encourage conversations, discussions, and mutual decisions on attributes that are strengths and areas for improvement that will be addressed in the action plan.

Sample interview questions and prompts are provided in File S2. We encourage involving the child in the discussion to the extent possible.

**Table 1.** Attributes specific to the participation goal.

| Child | Family | Environment |
|---|---|---|
| Interest and desire to participate in the activity | Interest and desire for the child to participate in the activity | Accessibility and safety of the place(s) where activity will occur |
| Knowledge and understanding of the activity | Daily routines and family structure related to the child's participation in the activity | Availability of physical assistance from peers and adults (non-family members) |
| Physical abilities (positioning, mobility, manual) | Concerns related to the child's participation in the activity | Availability of social-emotional support from peers and adults (non-family members) |
| Self-care (eating, dressing, hygiene/bathing, toileting) | Support for the child to participate in the activity | Availability of community resources |
| Communication abilities | Resources for the child to participate in the activity | Other |
| Social, emotional, and behavioral considerations | Other | |
| Sensory considerations | | |
| Health and safety considerations | | |
| Other | | |

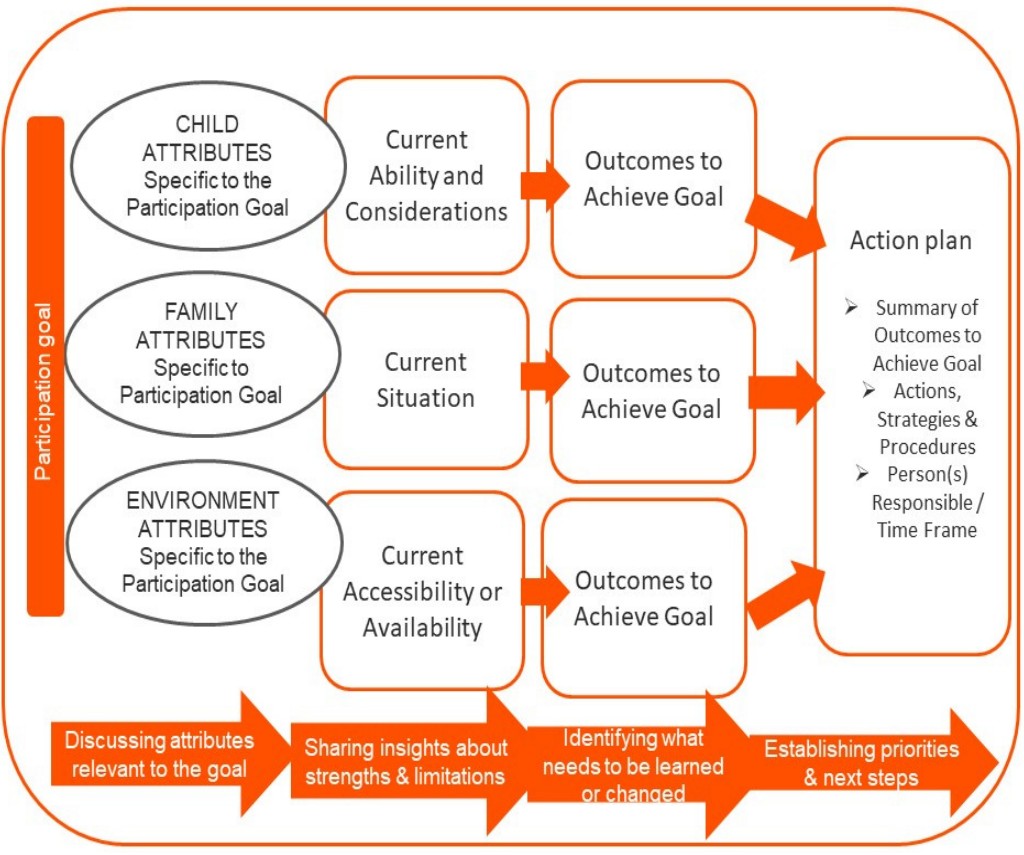

**Figure 2.** Flow diagram of Collaborative Process for Action Plans to Achieve Children's Participation Goals.

The action plan (Figure 2) is intended to highlight priorities, "first steps", and identify the person(s) responsible for initiating each action. The action plan might include strategies for communication and coordination, sharing information, education, instruction, identification and procurement of resources, and consultation. In many cases, responsibilities are shared but there may be some actions that the therapist takes primary responsibility and other actions that the family takes primary responsibility. The time frame for completion of the action plan is recorded. During implementation, therapists collaborate with the child, family, and community for a reciprocal exchange of information and insights and to learn from each other. The intent is for therapists to also share their expertise in a family-centered approach to build child, family, and community capacity. Sample interview questions and prompts for therapists to encourage conversations, discussions, and mutual decisions on the action plan are included in Table 2. Questions can be phrased for the child or parent. We encourage, to the extent possible, involving the child in the discussion.

**Table 2.** Questions for conversations, discussions, and mutual decisions for development of the action plan.

| |
|---|
| **To prioritize outcomes:** |
| • Let us discuss and decide together the most important areas for us to focus on to support your participation in the activity.<br>• What do you believe is the first outcome we should work on together?<br>• What outcome will be easy to accomplish?<br>• What outcome will take the most work and time to accomplish? |
| **To identify actions:** |
| • Let us discuss together some ways we can work on this outcome.<br>• What do you believe is the first action we should try together?<br>• What are your thoughts if we try ___________? |
| **To facilitate discussion on deciding who is responsible for the action:** |
| • What action do you feel comfortable working on?<br>• What action would you like my help with? |
| **To facilitate discussion on establishing the time frame for completing the actions:** |
| • How long do you think it will take to complete this action? |

The objectives of this international collaboration between researchers in the United States of America (USA) and Finland were to evaluate content validity and clinical utility of the Collaborative Process for Action Plans to Achieve Children's Participation Goals. Content validity is the degree to which items adequately reflect the content domain [37]. Clinical utility pertains to acceptability, feasibility, and usefulness [38]. In Phase I, content validity was examined in the USA using Nominal Group Consensus Process among therapists with expertise in pediatrics. Phase II involved translation to Finnish. In Phase III, clinical utility was evaluated in a cohort of children, families, and therapists in Finland. Final revisions were made by the authors in Phase IV. The findings should provide knowledge of strengths and limitations of a collaborative process to engage families and service providers in development of action plans to achieve participation goals and areas where further research is needed.

## 2. Phase I: Content Validity

### 2.1. Materials and Methods

Content validity was examined using Nominal Group Consensus Process. Nominal Group Consensus Process is intended to facilitate consensus among individuals with expertise in an area. The process involves presentation of a question, facilitation of structured

discussion, and voting after discussion [39]. Nominal group process typically involves five to nine participants. Two Nominal Group Consensus Process were conducted, the first in November 2010 and the second in August 2011. The second Nominal Group Consensus Process was conducted after revisions were made to the Collaborative Process for Action Plans based on the results of the first consensus process. The second Nominal Group Consensus Process also was conducted to include occupational therapists as participants.

### 2.1.1. Participants

The participants in the first Nominal Group Consensus Process were six physical therapists and one speech-language pathologist with 15 to 36 years of experience in pediatrics (mean: 23.3 years) who resided in one of three states in the USA (Pennsylvania, Delaware, Virginia). In this study, expertise was defined as a physical therapist, occupational therapist, or speech-language pathologist experienced in (a) providing services to children and families in community settings and (b) interventions for participation goals. Nineteen therapists, who were known to and judged by the first two authors as having the characteristics of an expert therapist were invited to participate; seven were able to participate.

Participants in the second Nominal Group Consensus Process were three occupational therapists and two physical therapists with 11 to 31 years of experience in pediatrics (mean: 23.8 years) who resided in one of three states in the USA (Pennsylvania, Oklahoma, Minnesota) or Ontario, Canada. Participants were selected from the list generated for the first Nominal Group Consensus Process and seven additional therapists identified to ensure representation of occupational therapists. Across both groups, therapists were from four different regions in North America and had leadership roles in early intervention, child development, school-based practice, hospital care, and academia. The Institutional Review Board at Drexel University determined that approval was not required because data were not being collected on human subjects.

### 2.1.2. Statements for Consensus

The statements used to evaluate content validity of the Collaborative Process for Action Plans are presented in Table 3. Following discussion, participants were asked to independently rate the extent they agreed that each child, family, and environment attribute was important for development of the action plan. Additionally, statements were included on whether knowledge of current performance/situation/accessibility and what needs to occur to achieve the participation goal are useful for development of the action plan. Clarity of wording also was rated. The final two statements address whether the Collaborative Process for Action Plans is easy to complete and provides important information for participation in home and community activities. Statements were rated on a five-point ordinal scale (Strongly Disagree, Disagree, Neither Agree nor Disagree, Agree, Strongly Agree).

### 2.1.3. Participant Preparation

Participants were e-mailed a letter that described the consensus process, the draft version of the Collaborative Process for Action Plans, and the questions for discussion. Participants were requested to: (a) become familiar with the content of the Collaborative Process for Action Plans prior to the teleconference, (b) think about a child and family they serve or had served and a goal for home or community participation when evaluating the Collaborative Process for Action Plans, and (c) write their thoughts as a prompt for discussion.

**Table 3.** Statements and mean ratings for nominal group consensus process.

| | Group 1 (n = 7) | Group 2 (n = 5) |
|---|---|---|
| **Child attributes** | | |
| Knowledge of the child's _________ is important for development of the action plan | | |
| (1) Interest in and motivation for the activity | 5.0 | 4.6 |
| (2) Knowledge and understanding of the activity | 4.7 | 4.0 |
| (3) Physical abilities pertinent to activity | 4.7 | 4.8 |
| (4) Communication abilities pertinent to activity | 4.9 | 4.8 |
| (5) Social, behavior and attention abilities | 4.7 | 4.8 |
| (6) Health and sensory issues pertinent to the activity | 4.6 | 4.6 |
| (7) The descriptions for child attributes are clear | 3.1 | 4.0 |
| **Family attributes** | | |
| Knowledge of the family's _________ is important for development of the action plan | | |
| (1) Attitude, interest, and perception of activity | 4.9 | 4.2 |
| (2) Daily routines and family structure pertinent to activity | 5.0 | 4.8 |
| (3) Safety concerns related to participation | 4.4 | 4.6 |
| (4) Awareness of the child's current abilities and modifications that may be required for participation | 4.7 | 4.2 |
| (5) Support for the child | 4.8 | 4.8 |
| (6) Resources for the activity | 4.9 | 4.6 |
| (7) The descriptions of the family attributes are clear | 3.6 | 4.2 |
| **Environment attributes** | | |
| Knowledge of ____________ is important for development of the action plan | | |
| (1) Physical accessibility | 5.0 | 4.2 |
| (2) Availability of physical support and assistance from peers and adults (non-family members) | 4.9 | 4.6 |
| (3) Social support from peers and adults (non-family members) | 5.0 | 4.6 |
| (4) Community resources | 4.9 | 4.2 |
| (5) The descriptions of environment attributes are clear | 4.4 | 4.4 |
| **Knowledge of ________is useful for development of the action plan** | | |
| (1) Current performance/situation/accessibility | 4.4 | 4.6 |
| (2) What needs to occur to achieve goal | 4.7 | 4.6 |
| **OVERALL**, The Collaborative Process for Action Plans is easy to complete and provides important information for participation in ________ | | |
| Home activities | 4.6 | 4.2 |
| Community activities | 4.6 | 4.2 |

Strongly Disagree = 1; Disagree = 2; Neither Agree nor Disagree =3; Agree = 4; Strongly Agree = 5.

### 2.1.4. Format of Nominal Group Consensus Process

The nominal group consensus process was conducted via teleconference. Two research assistants conducted the consensus meetings. One facilitated discussion and the other was timekeeper. The sessions began with a 5 min introduction. Twenty minutes was allotted for discussion of child, family, and environment attributes (total of 60 min). An additional 20 min was allotted for discussion of questions on relevance and usefulness. Following discussion of each question, participants independently rated each statement on the five-point ordinal scale and provided comments via an electronic survey. Following discussion of the last question, participants were encouraged to take a few minutes to go through the questions and provide any additional comments before submitting their responses.

### 2.1.5. Data Analysis

The data for each Nominal Group Consensus Process were analyzed separately. Consensus agreement for each statement was operationally defined by the authors as a rating of 'agree' (4) or 'strongly agree' (5) by at least 80% of participants and a mean rating $\geq 4.0$. Written comments were collated.

*2.2. Results*

In the first Nominal Group Consensus Process, ratings for 21 of 23 statements (91%) met the criterion for consensus agreement (Table 3). All participants 'agreed' or 'strongly agreed' that the Collaborative Process for Action Plans is easy to complete and provides important information. Consensus agreement was not achieved for clarity of the descriptions for child (mean rating: 3.1) and family (mean rating: 3.6) attributes. Revisions were made based on participant ratings and comments. A short introduction, a description of each attribute, sample interview questions, and the process for developing the action plan were added to improve clarity.

In the second nominal group process, all 23 statements (100%) met the criterion for consensus agreement (Table 3). No additional attributes were identified by the participants. Written comments were used to edit descriptions of attributes and interview questions. The conceptual framework for participation of children was added to the introduction and the format for recording the action plan was modified.

## 3. Phase II: Finnish Translation

Translation to Finnish and evaluation of clinical utility were carried out in 2016–2017 as part of the LOOK project on the Right of the Child to Participate in His/Her Rehabilitation in Helsinki, Finland. Two of the authors (NV and SS) translated the Collaborative Process for Action Plans to Achieve Children's Participation Goals to Finnish. The Finnish version was back translated into English by an official translator and reviewed by the first two authors. Small modifications were made to words whose meaning did not directly translate to Finnish.

A panel of 10 professionals, six occupational therapists, three physical therapists, and one music therapist/psychologist, read and provided comments via an electronic survey on the content, usefulness, feasibility, and clarity of the Finnish translation. The panel was recruited through LOOK-projects' meetings and email for the project's network that consisted of therapists and professionals who had special interest in children's participation-focused rehabilitation. Participation was voluntary. Panel members had 14–33 years (mean 22.4 years) of experience in pediatric rehabilitation, were knowledgeable of the International Classification of Functioning Disability and Health (ICF) [1], and were experienced in using GAS. Based on the panel's comments, minor edits were made to the wording of the Finnish version to improve clarity, decrease repetition, and highlight connection of attributes to the child's participation goal.

## 4. Phase III: Evaluation of Clinical Utility

*4.1. Materials and Methods*

4.1.1. Participants

Participants were a sample of convenience consisting of 10 family/therapist teams. Three teams included two pediatric therapists for a total of 13 pediatric therapists (6 occupational therapists, 6 physical therapists, 1 speech-language pathologist) with 4 to 40 years of experience (mean 17.6 years). One team included two parents for a total of 11 parents (2 fathers, 9 mothers). The 10 children with developmental delays or disabilities included 5 girls and 5 boys, 4–13 years of age (mean 7.9 years). Participants were recruited from three children's rehabilitation centers participating in the LOOK project. The therapists invited families on their caseload to partner with them to complete the Collaborative Process for Action Plans and provide feedback on acceptability, feasibility, and usefulness of the process. The Social Insurance Institution of Finland (funder) stated that ethics approval was not needed because the families are volunteer partners in developing the action plan and study questions pertain to the evaluation of the process.

4.1.2. Procedure

The process of family-therapist collaboration and interventions for participation goals were unfamiliar to most of the therapists and families. Therapists, therefore, received 2–5 h

of instruction on partnering with the child and family to complete the Collaborative Process for Action Plans. This included the sample questions and prompts to engage the child and family in the collaborative process.

The approach to goal setting used in the clinical settings where the therapists worked requires that the therapist and family (child and parents) set the goals for rehabilitation together. In this study, each of the 10 therapist-family teams was asked to select one participation goal prior to using the Collaborative Process for Actions Plans. The process of goal setting and the goal were not evaluated. Three goals were related to learning to ride a bike, two goals were related to going out in the community, two goals were related to recreation and leisure participation with peers, one goal was related to a home routine, one goal was related to self-care, and one goal was related to positioning.

Among the 10 family/therapist teams, five teams completed the Collaborative Process for Action Plans in a clinical setting, four in the family's home, and one at the child's daycare. Two professionals participated on three of the teams and two parents participated on one team. Time to complete the Collaborative Process for Action Plans varied from 30–90 min (mean: 76 min).

### 4.1.3. Feedback Questionnaires

Eight of 10 family/therapist teams jointly completed a feedback questionnaire after completing the form for the Collaborative Process for Action Plans. There were three main questions: (1) What did you like about the Collaborative Process for Action Plans? (2) What was challenging about the Collaborative Process for Action Plans? (3) What are the benefits of the Collaborative Process for Action Plans? The intent of the joint questionnaire was to enable the social construction of the knowledge by promoting open and collaborative reflection about the process by the family and therapist. Eleven therapists and eight parents then independently completed a separate feedback questionnaire to gain their individual perspectives. The individual questionnaire included open-ended questions on content and usefulness and a question about the child's engagement in the collaborative process.

### 4.1.4. Data Analysis

Participants' written responses to open-ended questions were translated into English and collated on one document for analysis. The transcripts were coded by NV and SS using inductive content analysis [40] to identify meaning units consisting of words, phrases, and sentences that represent similar content. Meaning units were organized into categories. All authors reviewed the transcripts, meaning units, categories, and arrived at consensus on strengths and limitations to evaluate clinical utility of the collaborative process.

### *4.2. Results*

The perceived strengths and limitations of the Collaborative Process for Action Plans that were identified from analysis of the open-ended responses on the questionnaires are presented in Table 4.

**Table 4.** Strengths and limitations of collaborative process for action plans to achieve participation goals perceived by parents and therapists.

| Strengths | Limitations |
| --- | --- |
| • Engagement of children, parents, and therapists<br>• Goals considered from many viewpoints<br>• Joint development and commitment to the action plan | • Long, laborious, and time consuming<br>• Complex process requires preparation and more guidance<br>• Unfamiliar expressions and words |

### 4.2.1. Strengths of the Collaborative Process for Action Plans

Engagement of children, parents, and therapists. Parents and therapists each indicated that engagement of the child and family in the discussion was a positive and useful aspect of the Collaborative Process for Action Plans and "all worked together". They noted that the process revealed the children's and families' concerns, desires, and interests; facilitated their understanding; and motivated them to work towards achieving the goal. Parents and therapists reported that conversations were "open", "informal", "jovial", and "fruitful". Therapists highlighted families' abilities to be analytical and collaborative.

Goal considered from many viewpoints. Therapists and parents expressed value in sharing perspectives about the goal. These discussions enabled a common, "concrete and practical" goal to be established together; "opened up the goal from different angles", which facilitated a better understanding and appreciation for the goal; and fostered a belief that the goal could be attained. Parents expressed that the Collaborative Process for Action Plans enabled reflection on the goal together and was useful in identifying meaningful details. One of the parents described that during the discussion there were new revelations regarding the child's feelings about the intended goal. In another instance it was noted that although the goal was expressed by the mother, the child realized why the goal was important to him. Therapists perceived that the Collaborative Process for Action Plans highlighted the family's contribution and the environment attributes as well as resources. Three therapists expressed an alternate viewpoint of the Collaborative Process for Action Plans having a narrow focus on one goal when there are several therapy goals. One therapist noted that when the goal did not come from the child it was hard in the beginning to engage the child.

Joint development and commitment to the action plan. Therapists and parents expressed that the resources and needs to achieve the goal were discussed from all perspectives. They shared that the attributes are relevant and the guiding questions useful to direct and deepen their conversations. Solutions were created and it became clear what was needed to achieve the goal. The Collaborative Process for Action Plans helped to focus on one activity at a time, and on the current situation with regard to the goal. A step by step jointly designed action plan that could be followed and revisited was considered valuable and motivating.

Decisions on the path to achieve the goal including collaboration with other providers, joint commitment to the action plan, and naming persons responsible for actions were perceived as benefits. Parents expressed that the Collaborative Process for Action Plans clarified the sharing of tasks and committed people to follow the plan. One parent/therapist team commented that "the process will help to substantiate goals and to commit all parties to more effectively work to reach them". Several parents indicated that discussions with therapists provided new ideas for practice of activities at home. Therapists commented that the Collaborative Process for Action Plans made it possible to bring up topics for discussion, immediately get to the point, establish clear rules for joint actions, and review content more concretely and with more diversification. Therapists highlighted that the Collaborative Process for Action Plans facilitated the identification of concerns related to a goal and environment attributes important for achieving a goal, that otherwise might have gone unnoticed.

### 4.2.2. Limitations of the Collaborative Process for Action Plans

Long, laborious, and time-consuming. Many parent/therapist teams noted that the process was "laborious", strenuous", and "burdensome". Some therapists expressed an alternate view that the process was not as laborious as they expected and became quicker with use.

Complex process requires preparation and more guidance. The participants noted that the Collaborative Process for Action Plans was challenging. One therapist stated, "it might be too hard for some families". Therapists and parents commented on the importance of a quiet environment for concentration. One therapist indicated that it was a "new way of

acting and conversations sometimes stumbled". Some parents and therapists expressed the desire for other adults who are part of the child's daily life to participate in the discussion. Therapists noted that they would like clearer instructions, tips for implementation including how to familiarize the parents with the collaborative process, and examples. Parents and therapists suggested that it would be beneficial to provide the parent with an introduction or the actual form before the visit so the "family could ponder things beforehand".

Unfamiliar expressions and words. Therapists and parents both expressed certain terms and questions were difficult to understand and answer due to unfamiliarity, especially related to environmental attributes and expressions for intervention planning. One parent stated that "she had to look for the instructions to understand some of the questions". A therapist stated that it "was challenging to lead the discussion and modify the questions and terms to be clear to the family at the same time". Some therapists and parents noted an alternate perspective that the forms were clear and easy to use.

### 4.3. Observation of Children's Participation in the Collaborative Process for Action Plans

Children's participation varied from enthusiastic and eager to answer questions and share aspirations to not being able to concentrate or avoiding unpleasant topics. In three cases the child was not present. One parent stated that the child's comments provided the family new insights. Another child was so enthusiastic that she imitated the collaborative process with her doll. Parents and therapists noted that some children were not able to participate fully in the process and were unable to express their opinion secondary to communication limitations. One parent preferred to discuss some of the family's concerns without the child present.

## 5. Phase IV: Final Revisions

Following Phase III, the authors met to consider final revisions. The following minor changes were made to the English version (File S1) and these minor changes were then translated into the Finnish version. Details were added to the instructions to emphasize that the process is flexible and can be adapted to individual contexts and needs. The descriptions of some attributes were modified in attempt to use words that are understandable and acceptable to families. Physical Abilities was changed to two separate child attributes: Physical Abilities (positioning, mobility, manual) and Self-care (eating, dressing, hygiene/bathing, toileting). Health and Sensory Issues was separated and reworded to Sensory Considerations and Health Considerations. We agreed to use the term 'action plan' throughout (previously 'intervention plan' was also used). Questions and prompts were added to assist in completion of the action plan. The phrase 'what needs to happen to achieve the goal' was changed to 'outcomes to achieve the participation goal'.

The Collaborative Process for Actions Plans to Achieve Children's Participation Goals including background information, instructions for completing, suggested open ended questions for discussions with families, examples of completed assessments, and a fillable form are available on the CanChild website https://canchild.ca/en/resources/335-the-collaborative-process-for-participation-goals (accessed on 22 June 2022). In Finland, a manual has been created to provide case examples and additional background information and strategies for therapists as well as information letter for the family to help them prepare for the collaborative process beforehand (http://www.theseus.fi/handle/10024/140228, accessed on 22 June 2022). An example of a completed form is provided in File S3.

## 6. Discussion

The results provide evidence of content validity of the Collaborative Process for Action Plans to Achieve Children's Participation Goals as well as insights to improve acceptability, feasibility, and usefulness. All phases of the study informed revisions to optimize clarity and ease of use in practice, including minor word changes, addition of examples, expansion of background information, and instructions to emphasize the flexibility and adaptability of the collaborative process. Evaluation of clinical utility in Finland highlighted that the

usefulness of the collaborative process may depend on therapists' and families' familiarity and support of practices associated with engagement of families in the therapy process. The feedback from therapists and families reflects the perspective that there are challenges to participatory practices [15–18] and individual differences in preferred practices. The findings extend knowledge of participatory practices to engage families in collaborative goal setting [27,28] to include participatory practices to engage families and therapists in development of action plans to achieve participation goals.

The strengths of the Collaborative Process for Action Plans (engagement of children, parents, and therapists; goal considered from many viewpoints; joint development and commitment to the action plan) are aligned with family centered and participation-based approaches to service delivery. Engagement, i.e., affective, cognitive, and behavioral involvement, and investment in the therapy process, is conceptualized as a means to realize family-centered care and optimize meaningful outcomes [11]. There was agreement on the usefulness and value of jointly establishing an action plan with clear responsibilities. The comprehensiveness of the process enabled the families and therapists to consider a range of child, family, and environment attributes to support participation. Together families and therapists were able to discuss, analyze, and create solutions.

The limitations of the Collaborative Process for Action Plans (long, laborious, and time-consuming; complex process requires preparation and more guidance; unfamiliar expressions and words) highlight the importance of therapist and family familiarity with solution focused therapy approaches and collaborative processes that are adaptable to individual preferences and resources. Most participants were not familiar with collaborative processes and interventions to achieve goals for participation. Hence, their perspective that the Collaborative Process for Action Plans is not for everyone. Our impression is that unfamiliarity with the process led some parent-therapist teams to discuss each child, family, and environment attribute rather than identifying the attributes pertinent to the participation goal and how they relate to each other. Similarly, parents and therapists previously may not have been asked to consider how attributes inform the action plan. Although the recommendation is to focus on a priority goal, the Collaborative Process for Actions Plans can be repeated to address additional participation goals, such as one for home/family life and one for the community recreation. However, the complexity and time requirements are important considerations. The finding that some participants expressed difficulty in understanding questions and maintaining discussion suggests that some therapists may benefit from instruction in strategies for engaging families.

Our findings support the importance of individual considerations, sensitivity, and flexibility in how to engage children and the use of children's preferred communication methods [41]. Franklin and Sloper [42] advocate that the child's participation in planning is a developing process that needs careful attention to child-centered practices and attitudes. Creative methods such as drawing and photographing [43] and adapting the Collaborative Process for Action Plans so that it is meaningful for the child have been useful for understanding children's views and engaging children as partners in the process [44]. In the LOOK-project, the CMAP (Child's Meaningful Activities and Participation) was developed to enable the child, with the help from an adult, to describe activities and participation that are important in everyday life with an application using photographs, videos, texts, recordings, and drawings [45].

Despite the focus on participation, some of the goals were not contextual and one was for body structures and functions. In Finland, the International Classification of Functioning, Disability and Health [1] has been in use to guide the assessment of children with disabilities, but clinical use of the ICF framework varies. The therapists participated in 2–5 h of instruction on the collaborative process. However, experiential learning and mentorship might be needed for use in practice. The manual with case examples and instructions developed in Phase IV incorporates feedback from parents, children, and therapists during Phase III to improve clinical utility. The limitations expressed by parents and therapists, in part, may reflect a mismatch between goals for activities that do not have an environ-

mental context and collaborative processes for action plans to achieve participation goals. Although our findings cannot be generalized, parents' and therapists' experiences illustrate the importance of cultural considerations and familiarity with collaborative processes.

*Limitations and Recommendations for Further Research*

A limitation of the Nominal Group Consensus Processes was not including parent participants. Clinical utility was evaluated in a small sample of 10 parent/therapist teams and findings are not generalizable. Although children are encouraged to actively participate in the collaborative process, this did not occur in all cases and children did not provide direct feedback. We did not measure socio-economic and environmental factors which could influence the collaborative process. To further determine acceptability, feasibility, and usefulness of the Collaborative Process for Action Plans, larger studies across varied geographical regions and practice settings are recommended in which the action plan is implemented, progressed, and achievement of participation goals are evaluated.

## 7. Conclusions

The Collaborative Process for Action Plans to Achieve Children's Participation Goals offers pediatric rehabilitation practitioners an approach to engage and proactively partner with children and families to assess, plan for, and evaluate supports and services to promote children's participation in activities that are meaningful to them. The findings of nominal group process provide evidence of content validity. Feedback from parents, children, and therapists who completed the collaborative process identified strengths (engages parents and therapists in conversations and discussions, goals are considered from many viewpoints, joint commitment to the action plan) and limitations (time needed to complete, complex process requires preparation and more guidance, unfamiliar expressions and words). The manual with case examples and instructions incorporates feedback from parents, children, and therapists to improve clinical utility. Familiarity with collaborative, solution focused approaches, and participation-based interventions are considerations for use in practice.

**Supplementary Materials:** The following supporting information can be downloaded at: https://www.mdpi.com/article/10.3390/disabilities2040045/s1, File S1. Form for Collaborative Process for Action Plans to Achieve Children's Participation Goals, File S2. Sample questions to assess child, family, and environment attributes, File S3. Example of Collaborative Process for Action Plans to Achieve Children's Participation Goals.

**Author Contributions:** Conceptualization, R.J.P., L.A.C., N.V. and S.S.; methodology, R.J.P., L.A.C., N.V. and S.S.; validation, R.J.P., L.A.C., N.V. and S.S.; formal analysis, R.J.P., L.A.C., N.V. and S.S.; investigation, R.J.P., L.A.C., N.V. and S.S.; resources, R.J.P., L.A.C., N.V. and S.S.; data curation, R.J.P., L.A.C., N.V. and S.S.; writing—original draft preparation, R.J.P., L.A.C., N.V. and S.S.; writing—review and editing, R.J.P., L.A.C., N.V. and S.S.; visualization, R.J.P., L.A.C., N.V. and S.S.; supervision, R.J.P., L.A.C., N.V. and S.S.; project administration, R.J.P., L.A.C., N.V. and S.S.; funding acquisition, N.V. and S.S. All authors have read and agreed to the published version of the manuscript.

**Funding:** The LOOK- project was managed by Metropolia University of Applied Sciences in collaboration with The Central Union for Child Welfare and funded by The Social Insurance Institution of Finland (Kela) (http://look.metropolia.fi/in-english/, accessed on 22 June 2022).

**Institutional Review Board Statement:** Content validity: The Institutional Review Board at Drexel University determined that approval was not required because data were not being collected on human subjects. Clinical utility: The Social Insurance Institution of Finland (funder) stated that ethics approval was not needed because the families are volunteer partners in developing the action plan and study questions pertain to the evaluation of the process.

**Informed Consent Statement:** Informed consent was obtained for all participants in Phase III—Evaluation of Clinical Utility.

**Data Availability Statement:** Not applicable.

**Acknowledgments:** We extend our thanks to the therapists and families involved in this study.

**Conflicts of Interest:** The authors declare no conflict of interest. The funder had no role in the design of the study; in the collection, analyses, or interpretation of data; in the writing of the manuscript, or in the decision to publish the results.

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
