# Peer review of "Content Validity and Utility of the Collaborative Process for Action Plans to Achieve Children’s Participation Goals"

_disabilities, doi:10.3390/disabilities2040045_

Round 1

Reviewer 1 Report

Comments

General comments

The topic under investigation is interesting, but lacking information in this report in the whole paper. Since there are a number of omissions in the reporting of this study that need to be addressed, the discussion of this study is questionable.

Abstract.

The abstract will need to re-write after finalizing the content of paper.

Introduction-content

Not read because method is very confusing.

Introduction-aim

The aim missing.

Methods: general

Method is missing. Authors mix up introduction and method.

Methods: design

No design available.

Readability and style.

This paper is not easy to read, because the information is unstructured

Author Response

We appreciate that the format of the manuscript is not conventional. The four-phase format was selected in an effort to report the entire study in a single article. Revisions have been made in an effort to improve clarity.

Reviewer 2 Report

This work addresses such an important topic, that is providing means for clinicians to foster a focus on achieving participation goals in pediatric rehabilitation. The steps of assessing content validity and clinical utility are the first steps in further evaluation to identify whether the Collaborative Process for Action Plans to Achieve Participation Goals is effective. The manuscript is very well written without awkwardness of expression, grammar etc.

In its current form, I don’t believe that this manuscript is suitable for publication. There are aspects of the methods in particular which don’t provide sufficient detail to enable appraisal of the rigour of the methods, or the ability to replicate this study. My aim is to provide information below to explain my view.

Title. The title clearly describes the topic/content but needs the participant group to whom the study is relevant (children/disability) and the design/purpose (content validity/clinical utility) articulated.

Abstract. This work has multiple steps and facets which must make it very difficult to adequately summarise the study aims, methods, findings and implications in the abstract. Due to this, I think, the abstract has gaps in its description of the study and in the clarity of the content. The first sentence is too long to comfortably and meaningfully read.

Introduction

The introduction provided an informative, logical background to the work and its importance.

For clarity consider changing to (around line 70): Collaborative processes, however, were unspecified or not well described. Similarly, the role of the child was unspecified or not well described. Only four studies described ongoing processes such as monitoring goal progress.

Lines 78-80: In 2 places it mentions that therapists interacted more with parents/children. I am not clear as to whether this is more than baseline or more than the other group.

Line 86: “Knowledge of participatory practices for development of action plans to achieve participation goals is limited”. After reading about the literature in the following sentences I am not clear what is meant by ‘limited’. Is there little work completed, or none; is the work inconclusive; are any studies low level? Can the state of the knowledge be made clear?

Line 87: Should Pathways and Resources for Engagement and Participation be mentioned amongst these interventions? I especially note a recent publication with children, see following. Note that I am not an author on the paper or involved in this research.

Killeen, H., & Anaby, D. R. (2022). The impact of parent involvement on improving participation of children born preterm: The story in the baseline. Contemp Clin Trials Commun, 28, 100942. https://doi.org/10.1016/j.conctc.2022.100942

Line 120 mentions that the participation goal is listed at the top of the form. I am not clear how the participation goal is identified or even defined. True participation goals are notoriously difficult to identify. It often requires a shift in thinking for therapist and child and family to develop truly participation goals. The COPM is often modified for the purpose but its focus on occupational performance does not readily adapt to participation goals without a skilled therapist guiding the process. In short, it would be very appropriate to discuss how participation goals are identified and defined. I am reminded of this again with the Finnish sample where it was stated that working in a participation frame of reference was unfamiliar. Further, when I read the goals used for Phase 4, I would argue that riding a bike is not a participation goal unless it is in a real life context, like riding a bike with family in central park for an hour. I also wonder how positioning is a participation goal. I imagine that a participation goal may be to go to the movies with friends but one barrier is pain during prolonged sitting which might be alleviated with better positioning. And now I read the discussion where you did also flag the issue of setting participation goals. This reinforces my views noted earlier in this paragraph about defining and identifying truly participation goals so that readers of this work are very clear about what they are and are not.

Line 127 “Some attributes may not apply for certain goals; for other goals, attributes not listed might be important”. Please make clear whether the “listed” means those things not listed on the form as prompts or not listed by children and families when responding to the questions on the form.

Line 268: “During implementation, therapists collaborate with the child, family, and community to share information, educate, and instruct in ways that build child, family, and community capacity.” As I read through table 2, my perception was that most, but not all, questions were asked to elicit knowledge for the therapist to then act upon. There were only some that appeared to draw on the parents’ problem solving and suggest collaborative processes. This is reinforced by the words noted above “share information, educate, and instruct”. The link between the questions and these words, and building “child, family, and community capacity” is not then clear. Is there a role for drawing on coaching as a strategy? Table 3 offers a clearly collaborative process

The tables which describe the Collaborative Process for Action Plans to Achieve Participation Goals, where they are currently situated in the manuscript are very distracting and provide too long a gap within the introduction section, meaning the continuity of this section is interrupted. Could these tables be consolidated in a single appendix or at the end of introduction section. In addition, given this study involved adapting a version of the Collaborative Process for Action Plans to Achieve Participation Goals, can it be made clear that the materials in the tables are the original or the adapted version.

Issues around participation. I realise that the foundational work on Palisano et al’s model was done 10 or more years ago, when the ICF was the main guiding framework around participation. Since then, Imms and colleagues have conceptualised the family of Participation Related Constructs. See reference below. Note, I am not an author on the paper nor involved in this work.

Imms, C., Granlund, M., Wilson, P. H., Steenbergen, B., Rosenbaum, P. L., & Gordon, A. M. (2017). Participation, both a means and an end: A conceptual analysis of processes and outcomes in childhood disability. Developmental Medicine and Child Neurology, 59(1), 16-25. https://doi.org/10.1111/dmcn.13237

There is such coherence between what I read in the tables about consideration of the environment and contexts (accommodation, accessibility, affordability etc) and aspects of the family of Participation Related Constructs. Please consider whether it would be worthwhile to align the two or to draw attention to an alignment.

When reading Tables 1 and 2, I am also strongly reminded of longstanding occupational therapy models, in particular the PEO-P model. Again, I have no investment in the following reference.

Baum, C. M., Christiansen, C. H., & Bass, J. D. (2015). The Person-Environment-Occupation-Performance (PEOP) Model. In C. H. Christiansen, C. M. Baum, & J. D. Bass (Eds.), Occupational therapy. Performance, participation, and well-being (4 ed., pp. 49-55). Slack Incorporated.

It may be useful to describe how your work is different to or adds value to such models – mainly to forestall other such commentary around the similarities.

The phases were logically presented and well written and represented a highly responsible series of steps in developing Collaborative Process for Action Plans to Achieve Participation Goals

The first paragraph 2.1 of the methods needs some amendment to achieve clarity, specifically to ground it in the context of this study. What were you planning to achieve consensus on (questions to include in the various sections of the Collaborative Process for Action Plans to Achieve Participation Goals  for instance)? It became clearer that you offered many questions on which to achieve questions, not the one question mentioned. Also here mention that two nominal group processes were completed, and the reasons for two. For instance, when reading the results section it appears that the Collaborative Process for Action Plans to Achieve Participation Goals was modified after the first group and the second group was a second round of consensus.  Mention also that they were via teleconference.

The time period during which each part of the study were conducted are necessary to include.

Table 4. Not all the statements in Table 1 are included in table 4. What is the reason for this? Perhaps they were added as part of the amendments following the first group?? If so, this can be specified in the reporting of the results.

Data analysis 2.1.5. Specify that the two groups were analysed separately. The source of your decision rules around consensus needs referencing. Does the sentence about written comments being transcribed need to be included, given they were already written. Perhaps the written comments were used to inform decisions about amendments to the questions?

The process of translation/back translation is responsible as is the evaluation by the expert panel. It sounds as though there were not cultural adaptation needs? It is not clear how/if these changes were integrated into the English version? A little information on how/from where these people were recruited is necessary. It would be worthwhile specifying that characteristics of this expert panel were not collected (e.g., age, years of experience especially with setting participation goals etc), or reported if they were.

Please include a brief overview of the methods for Phase 4, to give the reader a context eg therapist/families implemented the Collaborative Process for Action Plans to Achieve Participation Goals and gave feedback about…..

4.1.5: again, do you need to transcribe the written responses?

4.2. use of the word “emerged”. Some contemporary qualitative analysis gurus argue that researchers actively engage with narrative data and so identify or develop themes, rather than passively having them emerge.

4.2.3 is presented as if it is another theme, but it isn’t and needs a new number eg. 4.3 and perhaps a change of title to Observations of childrens ….

The clinical utility phase didn’t seem to identify any cultural adaption difficulties – were there any observations around this. It is clear that the Collaborative Process for Action Plans to Achieve Participation Goals was translated to Finnish for the clinical utility phase and amendments made as a result. Is the version in the manuscript (in the tables) the adapted version (as per comments above) and if so, describe the process to adapt and/or translate back to English or what methods were used to update the English version.

Discussion. This section raises really interesting points and brings together the findings in a way which illuminates the findings, their implications and recommendations for the future.

Line 534 used the term pilot testing. This is the first time this term is mentioned. The work reported is not pilot testing and best not to use the term. Pilot studies focus on outcomes, rather than process, and include an evaluation of participant responses to the intervention.

Section 6.1: “Implementation of the action plan and achievement of he participation goal were not evaluated”. I would advise not to include this sentence. This was clearly not the purpose of your study, rather you complete highly responsibility early steps to develop and feasibility test the Collaborative Process for Action Plans to Achieve Participation Goals AND you make the important point that the next steps are implementation and evaluation. Also you have specifically identified that evaluation of outcomes can be a next step. There are other options also: setting participation goals, fidelity of implementation of Collaborative Process for Action Plans to Achieve Participation Goals, evaluation of outcomes.

Author Response

Title. The title clearly describes the topic/content but needs the participant group to whom the study is relevant (children/disability) and the design/purpose (content validity/clinical utility) articulated.

Response: We have revised the title to read: Content Validity and Clinical Utility of the Collaborative Process for Action Plans to Achieve Children’s Participation Goals

Abstract. This work has multiple steps and facets which must make it very difficult to adequately summarise the study aims, methods, findings and implications in the abstract. Due to this, I think, the abstract has gaps in its description of the study and in the clarity of the content. The first sentence is too long to comfortably and meaningfully read.

Response: We are mindful that there is a 200-word limit. As recommended, we have shortened the first sentence.

For clarity consider changing to (around line 70): Collaborative processes, however, were unspecified or not well described. Similarly, the role of the child was unspecified or not well described. Only four studies described ongoing processes such as monitoring goal progress.

Response: We have revised the paragraph as recommended.  

Lines 78-80: In 2 places it mentions that therapists interacted more with parents/children. I am not clear as to whether this is more than baseline or more than the other group.

Response: We have revised the sentence to clarify that physical therapists who received instruction on implementation of the strategies interacted more with parents during therapy sessions compared with therapists in the comparison group.

Line 86: “Knowledge of participatory practices for development of action plans to achieve participation goals is limited”. After reading about the literature in the following sentences I am not clear what is meant by ‘limited’. Is there little work completed, or none; is the work inconclusive; are any studies low level? Can the state of the knowledge be made clear?

Response: We have revised the paragraph and added a paragraph in an effort to articulate what is unique about the Collaborative Process for Action Plans to Achieve Children’s Participation Goals and state that the process is compatible with contemporary approaches to children’s participation.

Line 87: Should Pathways and Resources for Engagement and Participation be mentioned amongst these interventions? I especially note a recent publication with children, see following. Note that I am not an author on the paper or involved in this research.

Response: We have added a reference for Pathways and Resources for Engagement and Participation which expands on the two references for environment based interventions.  

Line 120 mentions that the participation goal is listed at the top of the form. I am not clear how the participation goal is identified or even defined. True participation goals are notoriously difficult to identify. It often requires a shift in thinking for therapist and child and family to develop truly participation goals. The COPM is often modified for the purpose but its focus on occupational performance does not readily adapt to participation goals without a skilled therapist guiding the process. In short, it would be very appropriate to discuss how participation goals are identified and defined.

Response: We appreciate the comments. We struggled with how to emphasize that the Collaborative Process for Action Plans to Achieve Children’s Participation Goals is initiated after a participation goal is established. As approaches to collaborative goal-setting are available, our intent was to present an approach to collaborative discussions to develop an action plan. We appreciate that in practice goal setting and development of an action plan to achieve the goal are inter-related. In the Introduction, our approach is to summarize research on goal setting with emphasis on research on specific strategies for therapist-parent collaboration. In section 4.1.3 we have added information on the process for goal setting that the therapists who participated in evaluating clinical utility were familiar but indicate we did not evaluate goal setting.

Line 127 “Some attributes may not apply for certain goals; for other goals, attributes not listed might be important”. Please make clear whether the “listed” means those things not listed on the form as prompts or not listed by children and families when responding to the questions on the form.

Response: To clarify, we have indicated “attributes not listed on the form”.

Line 268: “During implementation, therapists collaborate with the child, family, and community to share information, educate, and instruct in ways that build child, family, and community capacity.” As I read through table 2, my perception was that most, but not all, questions were asked to elicit knowledge for the therapist to then act upon. There were only some that appeared to draw on the parents’ problem solving and suggest collaborative processes. This is reinforced by the words noted above “share information, educate, and instruct”. The link between the questions and these words, and building “child, family, and community capacity” is not then clear. Is there a role for drawing on coaching as a strategy? Table 3 offers a clearly collaborative process

Response: Thank you for your thoughtful insight. Our intent was to foster reciprocal processes.  We have edited our word choice and added that the sample interview questions and prompts are for therapists to encourage conversations, discussions, and mutual decisions on the action plan. We added the sentences “Questions can be phrased for the child or parent. We encourage, to the extent possible, involving the child in the discussion.”

The tables which describe the Collaborative Process for Action Plans to Achieve Participation Goals, where they are currently situated in the manuscript are very distracting and provide too long a gap within the introduction section, meaning the continuity of this section is interrupted. Could these tables be consolidated in a single appendix or at the end of introduction section. In addition, given this study involved adapting a version of the Collaborative Process for Action Plans to Achieve Participation Goals, can it be made clear that the materials in the tables are the original or the adapted version.

Response: We removed Tables 1 (the form) and Table 2 (sample questions to assesses child, family, and environment attributes) from the text. They are now Appendices. We added a Table that summarizes the Child, Family, and Environment attributes and a Figure (flow diagram) to illustrate the Collaborative Process for Action Plans to Achieve Children’s Participation Goals.

Issues around participation. I realise that the foundational work on Palisano et al’s model was done 10 or more years ago, when the ICF was the main guiding framework around participation. Since then, Imms and colleagues have conceptualised the family of Participation Related Constructs. See reference below. Note, I am not an author on the paper nor involved in this work.

There is such coherence between what I read in the tables about consideration of the environment and contexts (accommodation, accessibility, affordability etc) and aspects of the family of Participation Related Constructs. Please consider whether it would be worthwhile to align the two or to draw attention to an alignment.

When reading Tables 1 and 2, I am also strongly reminded of longstanding occupational therapy models, in particular the PEO-P model. Again, I have no investment in the following reference.

It may be useful to describe how your work is different to or adds value to such models – mainly to forestall other such commentary around the similarities.

Response: Thank you for the recommendations. We have added citations of the framework by Imms et al. (2017), the Person-Environment-Occupation-Performance Model, and the Pathways and Resources for Engagement and Participation.

The phases were logically presented and well written and represented a highly responsible series of steps in developing Collaborative Process for Action Plans to Achieve Participation Goals

The first paragraph 2.1 of the methods needs some amendment to achieve clarity, specifically to ground it in the context of this study. What were you planning to achieve consensus on (questions to include in the various sections of the Collaborative Process for Action Plans to Achieve Participation Goals for instance)? It became clearer that you offered many questions on which to achieve questions, not the one question mentioned.

Response: We have elaborated on the statements for the nominal group consensus process that were rated by participants to evaluate content validity in 2.1.2.

Also here mention that two nominal group processes were completed, and the reasons for two. For instance, when reading the results section it appears that the Collaborative Process for Action Plans to Achieve Participation Goals was modified after the first group and the second group was a second round of consensus.  Mention also that they were via teleconference.

Response: We have added rationale for two Nominal Group Consensus Process to 2.1.

The time period during which each part of the study were conducted are necessary to include.

Response: We have added the time period when data were collected. Phase I: Content Validity was completed between 2010-2011 and Phases II: Finnish Translation and III: Evaluation of Clinical Utility were completed between 2016-2017 following establishment of a partnership among authors from the United States and Finland.  With the opportunity to evaluate clinical utility we believed it was important to provider the reader with the findings related to content validity.

Table 4. Not all the statements in Table 1 are included in table 4. What is the reason for this? Perhaps they were added as part of the amendments following the first group?? If so, this can be specified in the reporting of the results.

Response: The statements in Table 4 (now Table 3) were discussed by participants in the two Nominal Group Consensus Process. Table 1 (now Appendix 1) is the form for the Collaborative Process for Action Plans to Achieve Children’s Participation Goals. As described in Phase V: Final Revisions, the descriptions of some attributes were modified based on participant comments in attempt to use words that are understandable and acceptable to families.

Data analysis 2.1.5. Specify that the two groups were analysed separately.

Response: We have added this information.

The source of your decision rules around consensus needs referencing.

Response: We have revised the sentence on the criterion for consensus to state: “Consensus agreement for each statement was operationally defined by the authors as a rating of ……” A literature search did not identify a criterion for consensus when agreement is rated based on five response options.

Does the sentence about written comments being transcribed need to be included, given they were already written. Perhaps the written comments were used to inform decisions about amendments to the questions?

Response: We have revised the sentence to state: “Written comments were collated.”

The process of translation/back translation is responsible as is the evaluation by the expert panel. It sounds as though there were not cultural adaptation needs? It is not clear how/if these changes were integrated into the English version?

Response: We have added the following two sentences to the end of the second paragraph in 3. Phase II: Finnish Translation. “These minor edits were not transferred to the English version. There were no comments pertaining to cultural adaptation difficulties.”

A little information on how/from where these people were recruited is necessary. It would be worthwhile specifying that characteristics of this expert panel were not collected (e.g., age, years of experience especially with setting participation goals etc), or reported if they were.

Response: We have added the following information. “The panel was recruited through LOOK-projects’ meetings and email for the project’s net-work that consisted of therapists and professionals who had special interest in children’s participation-focused rehabilitation. Participation was voluntary. Panel members had 14-33 years (mean = 22.4) of experience in pediatric rehabilitation between 14-33 years and were knowledgeable of the (mean 22,4) and with the International Classification of Functioning Disability and Health (ICF) [1] and GAS experience.”

Please include a brief overview of the methods for Phase 4, to give the reader a context eg therapist/families implemented the Collaborative Process for Action Plans to Achieve Participation Goals and gave feedback about…..

Response: We could not identify additional information to add.

4.1.5: again, do you need to transcribe the written responses?

Response: We revised the sentence to state: “Participants’ written responses to open-ended questions were translated into English and collated on one document for analysis.”

4.2. use of the word “emerged”. Some contemporary qualitative analysis gurus argue that researchers actively engage with narrative data and so identify or develop themes, rather than passively having them emerge.

Response: We concur and have revised the sentence to state: “The perceived strengths and limitations of the Collaborative Process for Action Plans that were identified from analysis of the open-ended responses on the questionnaires are presented in Table 4.”

4.2.3 is presented as if it is another theme, but it isn’t and needs a new number eg. 4.3 and perhaps a change of title to Observations of childrens ….

Response: We have made these two changes.

The clinical utility phase didn’t seem to identify any cultural adaption difficulties – were there any observations around this. It is clear that the Collaborative Process for Action Plans to Achieve Participation Goals was translated to Finnish for the clinical utility phase and amendments made as a result. Is the version in the manuscript (in the tables) the adapted version (as per comments above) and if so, describe the process to adapt and/or translate back to English or what methods were used to update the English version.

Response: In response to a comment by reviewer 2 added the following two sentences to the end of the second paragraph in 3. Phase II: Finnish Translation. “These minor edits were not transferred to the English version. There were no comments pertaining to cultural adaptation difficulties.”

In 5. PHASE IV: FINAL REVISIONS before describing the changes during the final revisions we added the following sentence: “The following minor changes were made to the English version (Appendix 1) and these minor changes were then translated into the Finnish version.”

Discussion. This section raises really interesting points and brings together the findings in a way which illuminates the findings, their implications and recommendations for the future.

Line 534 used the term pilot testing. This is the first time this term is mentioned. The work reported is not pilot testing and best not to use the term. Pilot studies focus on outcomes, rather than process, and include an evaluation of participant responses to the intervention.

Response: We have revised the sentence to state: “Evaluation of clinical utility in Finland….”

Section 6.1: “Implementation of the action plan and achievement of the participation goal were not evaluated”. I would advise not to include this sentence. This was clearly not the purpose of your study, rather you complete highly responsibility early steps to develop and feasibility test the Collaborative Process for Action Plans to Achieve Participation Goals AND you make the important point that the next steps are implementation and evaluation. Also you have specifically identified that evaluation of outcomes can be a next step. There are other options also: setting participation goals, fidelity of implementation of Collaborative Process for Action Plans to Achieve Participation Goals, evaluation of outcomes.

Response: We removed the sentence.  

Reviewer 3 Report

This is a well-written paper that addresses a timely topic: how to increase participation in children to improve their well-being. The specific focus is on children with motor disability (e.g., children trying to improve a home routine). But the same challenges of child/family participation are likely to hamper learning, compliance, and adherence in education and health centers more generally. The authors propose an interview tool that therapist could be used with families to improve child/family participation. Data pertain to survey results with experts (to establish content validity) and interviews with family/therapist groups (to use input to improve the tool).

I am very supporting of seeing this work published. Below are suggestions to improve the paper.

1.     The authors focus a great deal on the idea of child/family participation, but relatively little on the idea of getting this participation in a context of goal-setting. The literature on goal-setting is vast. I encourage the authors to include some of this literature (beyond merely stating that it exists) to explain why the proposed tool relies so heavily on goal-setting. What is the rationale of including goal-setting? Ideally, the authors present findings that establish univocally the relation between goal-setting and participation.

2.     It is not clear how the proposed tool relates to what already exists. To what extend are specific participatory practices needed to engage families? How, for example, does the proposed 'Collaborative Process for Action Plans to Achieve Participation Goals' expand on Palisano et al.’s model for participation-based therapy (if at all)? And how about its relation to Occupational Performance Coaching; Context Therapy; Environmental Based Intervention; or Adapted Community Events, all of which advocate for family-therapist collaboration. It might help to have a table comparing all these different therapies in what they offer and what is missing that justifies the need for the proposed tool. The table should make it clear how the proposed form (1) builds upon, (2) improves on, and (3) deviates from already existing tools.

3.     While well-written, the manuscript features some jargon that slows down readers who might be less familiar with the field. For example, what is a/the Nominal Group Consensus Process? Along the same lines, it would help to be consistent about how the proposed tool is referred to. Sometimes its name is italicized, but not always. And it is referred to as ‘Collaborative Process for Action Plans to Achieve Participation Goals’, or merely as ‘collaborative process’, or as ‘process of family-therapist collaboration and interventions for participation goals’. Consistency would benefit the readers greatly.

4.     In the participant section, it is confusing to describe the therapists, parents, and children separately from each other and separately from the settings used and the goals they worked on. Was same child was treated by more than one therapist? Did one child have two parents participating? Did two children pursue the same goal? Better would be to use the family/therapist team as unit to describe the participants, settings, and goals. I gather there were 10 family/therapist teams. That would be easier to understand than what is currently in the participant section (that there are 13 therapists, 11 parents, 10 children, 10 settings, and 9 goals).

Author Response

The authors focus a great deal on the idea of child/family participation, but relatively little on the idea of getting this participation in a context of goal-setting. The literature on goal-setting is vast. I encourage the authors to include some of this literature (beyond merely stating that it exists) to explain why the proposed tool relies so heavily on goal-setting. What is the rationale of including goal-setting? Ideally, the authors present findings that establish univocally the relation between goal-setting and participation.

Response: We received a similar comment from Reviewer 2. As previously stated, we struggled with how to emphasize that the Collaborative Process for Action Plans to Achieve Children’s Participation Goals is initiated after a participation goal is established. As approaches to collaborative goal-setting are available, our intent was to present an approach to collaborative discussions to develop an action plan. We appreciate that in practice goal setting and development of an action plan to achieve the goal are inter-related. In the Introduction, our approach is to summarize research on goal setting with emphasis on research on specific strategies for therapist-parent collaboration. In section 4.1.3 we have added information on the process for goal setting that the therapists who participated in evaluating clinical utility were familiar but indicate we did not evaluate goal setting.

It is not clear how the proposed tool relates to what already exists. To what extend are specific participatory practices needed to engage families? How, for example, does the proposed 'Collaborative Process for Action Plans to Achieve Participation Goals' expand on Palisano et al.’s model for participation-based therapy (if at all)? And how about its relation to Occupational Performance Coaching; Context Therapy; Environmental Based Intervention; or Adapted Community Events, all of which advocate for family-therapist collaboration. It might help to have a table comparing all these different therapies in what they offer and what is missing that justifies the need for the proposed tool. The table should make it clear how the proposed form (1) builds upon, (2) improves on, and (3) deviates from already existing tools.

Response: Thank you for the comments. We have revised the Introduction in an effort to indicate that the Collaborative Process for Action Plans to Achieve Children’s Participation Goals is not an intervention approach per se but rather is intended to provide suggested guiding questions to engage the child, family, and others in conversations for planning an intervention to achieve a participation goal.  We have added that the process is adaptable for client and practice contexts and compatible with other contemporary approaches to children’s participation.

While well-written, the manuscript features some jargon that slows down readers who might be less familiar with the field. For example, what is a/the Nominal Group Consensus Process?

Response: We appreciate the importance of describing procedures that might not be familiar to readers.

In 2.1 we state: Nominal Group Consensus Process is intended to facilitate consensus among individuals with expertise in an area. The process involves presentation of a question, facilitation of structured discussion, and voting after discussion [37]. Nominal group process typically involves five to nine participants.

The questions for discussion are included in Table 3.

We have revised 2.5 to state:  Consensus agreement for each statement was operationally defined by the authors as a rating of ‘agree’ (4) or ‘strongly agree’ (5) by at least 80% of participants and a mean rating ≥ 4.0. 

We would be happy to elaborate further on the Nominal Group Consensus Process if specific recommendations are provided.  

Along the same lines, it would help to be consistent about how the proposed tool is referred to. Sometimes its name is italicized, but not always. And it is referred to as ‘Collaborative Process for Action Plans to Achieve Participation Goals’, or merely as ‘collaborative process’, or as ‘process of family-therapist collaboration and interventions for participation goals’. Consistency would benefit the readers greatly.

Response: We have revised the manuscript to address this comment. In the Introduction, we indicate that the Collaborative Process for Action Plans to Achieve Children’s Participation Goals is referred to as Collaborative Process for Action Plans throughout the manuscript. We have edited the manuscript accordingly.  

In the participant section, it is confusing to describe the therapists, parents, and children separately from each other and separately from the settings used and the goals they worked on. Was same child was treated by more than one therapist? Did one child have two parents participating? Did two children pursue the same goal? Better would be to use the family/therapist team as unit to describe the participants, settings, and goals. I gather there were 10 family/therapist teams. That would be easier to understand than what is currently in the participant section (that there are 13 therapists, 11 parents, 10 children, 10 settings, and 9 goals).

Response: Added a topic sentence to 4.1.2 to indicate “Participants were a sample of convenience consisting of 10 family/therapist teams.” We then indicate 3 teams included 2 therapists (total of 13) and one team included two parents (total of 11).

Reviewer 4 Report

This is an interesting paper, regarding the description of goals not only by therapists, but also by parents and children. The full engagement of the patients and their families is a key factor in the progress. The protocols enabling the creation/definition of goals are giving the tools and nececessary structure.

I have only two concerns with this paper:

- limited numer of patients who participated in the final assessment of the protocol, and that only girls were invited. Despite the high equality of sexes in Finland (probably one of the highest in the world) still some differences between sexes regarding the goals could exist. 

- there was no discussion about the socio-economic and envronmental factors (living in town, living in the countyside) which could influence the goal defining proces.

Author Response

- limited numer of patients who participated in the final assessment of the protocol, and that only girls were invited. Despite the high equality of sexes in Finland (probably one of the highest in the world) still some differences between sexes regarding the goals could exist. 

Response: The sentence (4.1.2) “10 children with developmental delays or disabilities (5 girls, 4-13 years of age, mean 7.9 years)” is the source of confusion. We have revised the sentence to indicate “5 girls and 5 boys.”

- there was no discussion about the socio-economic and envronmental factors (living in town, living in the countyside) which could influence the goal defining proces.

 Response: We have added this to the limitations of the study.